# Automatic Change Detection System over Unmanned Aerial Vehicle Video Sequences Based on Convolutional Neural Networks

**DOI:** 10.3390/s19204484

**Published:** 2019-10-16

**Authors:** Víctor García Rubio, Juan Antonio Rodrigo Ferrán, Jose Manuel Menéndez García, Nuria Sánchez Almodóvar, José María Lalueza Mayordomo, Federico Álvarez

**Affiliations:** 1Visiona Ingeniería de Proyectos S.L., 28020 Madrid, Spain; nsanchez@visiona-ip.es (N.S.A.); jlalueza@visiona-ip.es (J.M.L.M.); 2Grupo de Aplicación de Telecomunicaciones Visuales, Escuela Técnica Superior de Ingenieros de Telecomunicación, Universidad Politénica de Madrid, 28040 Madrid, Spain; jrf@gatv.ssr.upm.es (J.A.R.F.); jmm@gatv.ssr.upm.es (J.M.M.G.); fag@gatv.ssr.upm.es (F.Á.)

**Keywords:** change detection, convolutional neural networks, moving camera, image alignment, UAV

## Abstract

In recent years, the use of unmanned aerial vehicles (UAVs) for surveillance tasks has increased considerably. This technology provides a versatile and innovative approach to the field. However, the automation of tasks such as object recognition or change detection usually requires image processing techniques. In this paper we present a system for change detection in video sequences acquired by moving cameras. It is based on the combination of image alignment techniques with a deep learning model based on convolutional neural networks (CNNs). This approach covers two important topics. Firstly, the capability of our system to be adaptable to variations in the UAV flight. In particular, the difference of height between flights, and a slight modification of the camera’s position or movement of the UAV because of natural conditions such as the effect of wind. These modifications can be produced by multiple factors, such as weather conditions, security requirements or human errors. Secondly, the precision of our model to detect changes in diverse environments, which has been compared with state-of-the-art methods in change detection. This has been measured using the Change Detection 2014 dataset, which provides a selection of labelled images from different scenarios for training change detection algorithms. We have used images from dynamic background, intermittent object motion and bad weather sections. These sections have been selected to test our algorithm’s robustness to changes in the background, as in real flight conditions. Our system provides a precise solution for these scenarios, as the mean F-measure score from the image analysis surpasses 97%, and a significant precision in the intermittent object motion category, where the score is above 99%.

## 1. Introduction

The use of change detection algorithms is crucial in high precision surveillance systems. The methods that make use of those algorithms aim to detect the differences between information acquired at the same location, e.g., an image captured in different moments. Unmanned aerial vehicles (UAVs) became a revolution in the surveillance sector due to the lower cost and reduced human workload needed compared to previous systems. In addition, UAV operations can be automatized. This need of automation increases the importance of change detection methods. These methods are based on image sequences analysis, usually acquired by mobile vehicles. Image acquisition from the mentioned vehicles entails a considerable issue for change detection algorithms: The camera movement. This is the fundamental challenge of the algorithms, as the movement produces a variable background, thus the flight’s route will be moderately modified from one flight to another. Furthermore, the weather conditions and the precision of GPS positioning influence the relation between the acquired frames and the location of the UAV. Compared to moving cameras, stationary cameras significantly reduce the complexity of the change detection problem, as the background is common to every image [1].

Moving cameras introduce complexity to the problem because the reference image is continuously changing. As a result, the system needs to detect the background of each image to provide precise detection. Therefore, variable backgrounds introduce a considerable computational load. This generates a complex problem to solve for real-time video surveillance tasks. An interesting approach for change detection with moving cameras is studied in [2]. Our method is based on reconstruction techniques. However, the reconstruction process alone would not be precise enough for our task. This is a consequence of the limited precision of CNNs to generate detailed images. In addition, it conforms an additional process that could slow down the system. Nevertheless, it provides a distinct perspective from the stationary camera algorithm mentioned above. After reviewing the different state-of-the-art approaches such as [2,3,4], we have observed an increasing tendency to use CNNs on image processing systems for change detection. Furthermore, studied implementations do not consider a moving camera, as in [3,5]. Consequently, a supplementary component has been considered to overcome the problems introduced by a variable background. Therefore, we have developed our own complete approach based on image alignment and CNNs.

In this document, we have reviewed the state-of-the-art implementations for change detection and image feature extraction in Section 2. Section 3 includes a detailed explanation of the process followed to develop our implementation. Section 4 describes the metrics obtained by our algorithm and compares them with the state-of-the-art implementations analysed previously. Finally, in Section 6, the most relevant inferences obtained from the development are explained.

## 2. Related Work

In this section, a review of traditional change detection is performed in Section 2.1. In Section 2.2, an explanation of convolutional neural networks is presented. In Section 2.3, we provide an analysis of change detection implementations based on convolutional neural networks. Finally, in Section 2.4, different traditional image processing techniques to extract and find common features between images are reviewed.

### 2.1. Change Detection Using Traditional Image Processing

Traditional image processing methods for change detection are based on pixel intensity variation modelling. This is the prime feature of most background subtraction algorithms. Implementations such as the Gaussian mixture model [6] are exceptionally popular among them. This method uses multiple Gaussian distributions to model the intensity value of each pixel to differentiate between background and foreground elements. However, this method does not perform efficiently in complex environments where movement is permanent. Other relevant approaches are based on kernel density estimation (KDE) [7]. This method also employs probability density functions to model the intensity of background pixels. Based on the mentioned distributions, the foreground pixels are detected. These methods are founded on the intensity regions to estimate background probability. Most of them are unable to model frequent events. They also adapt poorly to dynamic background situations. Modern approaches in terms of change detection are SuBSENSE [8] and ViBe [9]. SuBSENSE relies on spatio-temporal binary features as well as colour information to detect changes. In case of ViBe, for each pixel, a set of values taken in the past at the same location is acquired. This set is compared to the current pixel value to determine if it belongs to the background. After that, the model is adapted by choosing randomly which values to substitute from the background model. Both methods are based on elemental features, such as colour analysis. The lack of high-level information brings a decrease in performance in complex environments, which represent our principal target.

### 2.2. Convolutional Neural Networks

CNNs applied to image analysis usually constrain their input to remain as a three-dimensional element with the following parameters: Width, height and colour channels (depth). The network is conformed by three types of layers: Convolutional, activation and pooling. The task of each layer is:Convolutional layers perform a dot product between each region of the input and a filter. These filters, also known as kernels, define the size of the mentioned regions. With this operation, the layer is capable of obtaining the most significant features of the input. The number of filters typically varies from each convolutional layer, in order to obtain higher or lower level features. The resultant structure depends on the padding employed on the convolution.Activation layers execute a predefined function to their inputs. The aim of these layers is to characterize the features obtained from the previous layers to obtain the most relevant ones. This process is performed using diverse functions, which may vary among the systems.Pooling layers down-sample their input, obtaining the most significant values from each region. Most CNNs employ a 2 × 2 kernel in their pooling layers. With this region size, the input is down-sampled by half in both dimensions, width and height.

The concatenation of these structures allows the system to learn different level features and generate a more precise prediction than traditional methods, based on low-level features such as colour. Therefore, the application of convolutional neural networks has supposed a revolution in image processing. A more in-depth explanation of these neural networks can be found in [10].

### 2.3. CNN-Based Change Detection

As mentioned in the introductory section, researchers have developed methods to detect changes in images based on CNN architectures. Two methods have been studied and analysed to obtain insights about the architectures and accuracy levels that form the state-of-the-art nowadays. In [3], both background and foreground images are sliced in patches with reduced dimensions. The two patches (back and foreground) are concatenated in the depth axis, obtaining a unique structure to feed the CNN as a result. Convolution applied to images stacked depth-wise produces faster and more appropriate results than in any other dimension as detailed in [11]. As the result of introducing the concatenated structure to the CNN-based architecture, this model returns a black and white binary patch, with the detected foreground zones in white and background zones in black. Nevertheless, the restriction of using a moving camera is unsolved as the system is indicated for static acquisition devices. In [12], a variation of the Visual Geometry Group Net (VGGNet) architecture is connected to a deconvolutional neural network. VGGnet is a state-of-the-art implementation developed by VGG (Visual Geometry Group) from the University of Oxford [13]. The resultant architecture is applied to obtain an image of the same dimensions as the input picture. To do so, the number of deconvolution layers is equivalent to the CNN layers in the VGGNet. As a result, the model returns a binary image with identical size to the input image. The stated method does not employ a background image. It employs only the foreground and, for training purposes, the ground-truth images. As a result, we can retrieve two conclusions: First, without a background reference, the system can be employed for moving cameras if the dataset is properly conformed; next, the algorithm has to learn a different background for each scene. Therefore, the system will have to learn on each scene, which slows processing time in real-world applications. From the results of both methods, it may be concluded that CNNs have improved considerably the accuracy of traditional change detection algorithms.

### 2.4. Image Matching Techniques

On stationary camera situations, acquiring a precise reference image represents an effortless task, while on moving camera scenarios it remains a challenging problem. One of the recent progresses of UAV technology consists on the automation of flights [14]. This system provides human–UAV interaction by allowing the programming of UAV’s routes, specifying the coordinates at each point and other parameters, namely speed, height, mode of flight or behaviour, in case of any event possible. However, the mentioned innovation is subject to the real-world conditions that affect the behaviour of UAVs. As explained in Section 1, the image acquisition is assumed to vary from two flights on the same route. It can be caused by multiple reasons, in particular, modifications in UAV’s height, position because of GPS precision variations, wind conditions or movement of the camera. Consequently, the reference pictures obtained on one of those flights are not circumscribed to exactly the same area. In this paper we have selected an approach based on ORB (oriented FAST and rotated BRIEF) algorithm [15] to resolve the alignment problem. ORB descriptor is based on BRIEF (binary robust independent elementary features) [16] and FAST (features from accelerated segment test) [17] algorithms. This descriptor extracts the key points of an image by implementing FAST method. It employs a Harris corner measure [18] to discover the optimal points. To obtain the orientation Equation (Equation 3), ORB uses a rotation matrix based on the computation of intensity centroid [19], Equation (Equation 2) from the moments Equation (Equation 1). Moments are obtained using:(1)$$m_{pq}=\sum_{x,y}x^{p}y^{q}I\left(x,y\right),$$
where I(x,y) represents pixel intensities, *x* and *y* denote the co-ordinates of the pixel and finally, *q* and *p* indicate the order of moments. The centroid is obtained from the calculated moments as:(2)$$C=(\frac{m_{10}}{m_{00}}\;\frac{m_{01}}{m_{00}})$$

Finally, a vector from corner’s center to the centroid is constructed. The orientation is:(3)$$\theta \;=atan2(m_{01},m_{10})$$
where atan2 is the arctangent’s quadrant-aware version given by Equation (Equation 4):(4)$$atan2\left(x,y\right)=\left\{\begin{array}{cc} arctan(y\cdot x^{-1}) & \;x>0,\\ arctan(y\cdot x^{-1}+\pi ) & \;y\geq 0,x<0,\\ arctan(y\cdot x^{-1}-\pi ) & \;y<0,x>0,\\ \pi /2 & \;y>0,x=0,\\ -\pi /2 & \;y>0,x=0,\\ undefined & \;x=y=0. \end{array}\right.$$

Other descriptors considered during the development have been SIFT (scale-invariant feature transform) [20] and SURF (Speeded-up robust features) [21] algorithms. SIFT algorithm transforms a picture into feature vectors. Each vector is invariant to image translation, scaling, and rotation. After that, it compares each vector from the new image to the ones obtained from the reference picture. Next, it provides a candidate by matching features based on Euclidean distance. SURF algorithm obtains the feature vectors based on the sum of the Haar wavelet response around the points of interest. These points have been detected previously by using an integer approximation of the determinant of the Hessian matrix. However, the ORB descriptor is able to perform two orders of magnitude faster than SIFT and several times faster than SURF. Because of this performance difference, we have selected ORB to process our videos and reduce the computational cost of the system.

## 3. Methodology

Image alignment techniques employed on the approach are explained in Section 3.1. After that, the architecture of the deep neural network model is explained in detail in Section 3.3. The dataset generation for the process has been described in Section 3.4. Subsequently, the training process is detailed in Section 3.5. Lastly, the post-processing part is described in Section 3.6 to introduce how we have obtained our desired image from the model output. In Figure 1, a block diagram of a system is shown to detail our pipeline.

### 3.1. Image Alignment

As stated on previous sections, our system is applied to moving cameras. Our objective is to compare two video sequences (background and foreground) to detect the changed regions between them. Because of implementation purposes, a reference video from the UAV’s route is required, which will be considered as background. More recent videos from the same route are considered as our foreground scenarios. In real-world situations, the new images will not be completely aligned to the reference video. This can be caused by multiple reasons such as lack of GPS precision or weather variations. To solve the problem, an image alignment system has been developed using ORB [15]. The idea is similar to the feature alignment performed in [22]. Both the reference and the new route’s images are compared. Feature extraction is performed with ORB algorithm to obtain the most significant zones of each picture. After that, a descriptor matcher algorithm from [23] is created. The descriptor performs an analysis of the obtained values and outputs the relation between them by distance difference. This output is filtered and sorted to obtain the most adjacent descriptive points of both images. Lastly, a geometric transformation is performed to generate a modified version of the acquired image, aligned to the reference. An example of the results obtained by this system is depicted in Figure 2. As can be observed, the resultant image contains a black zone that represents the pixels from the foreground image not included in the reference. To prevent the appearance of false positives because of these black zones, only the mid section of the image is selected automatically to perform the change detection. The alignment system entails an innovation to other implementations such as [3,12], which consider a scenario with a static camera.

### 3.2. Sliding Window

The image provided to our system can vary in size depending primarily on the UAV’s camera device or time processing requirements. For instance, the processing requirements of a real-time detection system varies from those of a post-flight analysis. Moreover, deep learning models usually struggle to work with very high-resolution images. Our approach to resolve these problems is described in this subsection. In the first place, the maximum input size is defined as a parameter of the system. If the input images overcome the defined dimensions, they are resized to the specified size. After that, we have to divide the pictures into small regions. This is a consequence our network’s requirements to process images in reasonable time. Therefore, we can obtain pixel-level precision results.

To do so, we have employed an algorithm named sliding window. This algorithm iterates over the image’s dimensions, retrieving a matrix of a specific size (window) which contains a region of the initial image. The dimensions of the window are identical in width and height to prevent any further complication of the segmentation process. Another parameter of the algorithm is defined as the step. The step of a sliding window algorithm represents the distance, in each dimension, from the starting point of a window in iteration i to the starting point of the region in iteration i + 1. Adaptive to each dimension, the step remains a crucial factor in terms of computational cost, just as the dimension of the window. With these two parameters we can control the overlapping of zones among adjacent windows. If overlapping occurs, the system will benefit from an increment of precision on the analysis. In this case, four predictions are generated for the most part of the image, except the limits of each dimension. As a conclusion, the variation of both parameters provides an extremely efficient tool that can modify the algorithm’s performance to adapt it for multiple purposes: real-time segmentation, maximum precision prediction or sample generation. The application of this algorithm in both the reference image and background image is depicted in Figure 3.

### 3.3. Deep Neural Network Architecture

The model is based on the concatenation of two input images: The reference background scene and the updated scene image which may contain some changes. Both images are merged in depth dimensions, as it is performed in other state-of-the-art methods such as [3,4,12]. This input form allows the model to learn hierarchical features. As a result, CNNs conform an effective tool to obtain relevant information from images. An exhaustive description of the CNN architecture is described in detail in Table 1 and Table 2. Figure 4 illustrates the complete architecture. The deep neural network is composed by four CNNs with increasing number of filters and a kernel size of 3 × 3 pixels. As our inputs consist of images with reduced dimensions, we employ this kernel size to extract the features as detailed as possible to obtain a precise detection. For the activation layers of the CNN, we have used Rectified Linear Unit (ReLU) activation [24]. ReLU activation applies Equation (Equation 5) to its inputs. This function is widely implemented in CNNs as mentioned in Section 2.2 because of its reduced computational cost and the acceleration of the optimization process. Following this layer, we have employed a max-pooling layer with 2 × 2 kernel size.
(5)R(z)=max(0,z)

After the last CNN structure, we have vectorized its output neurons. The first one-dimensional layer contains a dropout layer [25] to avoid overfitting [26]. Continuing the structure, we have employed a batch normalization [27], which subtracts the mean of its inputs and divides them by the standard deviation as in Equation (Equation 6):(6)$$\hat{x_{i}}=\frac{x_{i}-\mu_{B}}{\sqrt{\sigma_{B}^{2}+\epsilon }}$$
denoting $\mu_{B}$ as the average value of the batch, $\sigma_{B}$ as the standard deviation of the batch, and ϵ a constant added for numerical stability. The dropout layer deactivates part of the neurons from our densely connected layer during the training process. As a result, the model improves its generalization, as it forces the layer to predict the same output using different neurons. The aim of batch normalization is to increase the stability of the model by normalizing the output of the previous layer. As a consequence, the model will be adaptable to new scenarios, which is one of the essential features of the proposed system. As our output consists of pixels from changed regions or pixels from unchanged regions, we are in a binary classification problem. Therefore, the final output layer uses a sigmoid activation function as depicted in Equation (Equation 7):(7)$$\sigma \left(z\right)=\frac{1}{1+e^{-z}}$$

The resulting normalized vector is the initial prediction of our system. It conforms the input of the post-processing methods described in Section 3.6. From this vector we construct a one-channel image that represents the changes between the reference image and the updated image.

### 3.4. Dataset

To train our model, we have selected images from CD2014 dataset [1]. This dataset contains images categorized in: “Baseline”, “Dynamic Background”, “Camera Jitter”, “Intermittent Object Motion”, “Shadows”, “Thermal”, “Bad Weather”, “Low Framerate”, “PTZ”, “Night” and “Air Turbulence”. We have picked “Bad Weather”, “Dynamic Background” and “Intermittent Object Motion” for our training process. This election has been based on the similarity of these images with real conditions where our model is to be applied. As stated before, the target of this system is to perform change detection on images obtained with variable weather and with dynamic scenarios. Therefore, the mentioned categories represent a stable base for our model to learn.

To build our dataset, we have selected one background image from each category to conform the background for each foreground picture. This has been done to provide a unique reference for the complete dataset and prepare the model for slight changes as mentioned in previous sections. To do so, each background image has been replicated to obtain a reference for each foreground picture. Because of computational limitations, the input size for our model represents a restrictive value. Two complete images could not be established as an input. Therefore, a division of each image must be performed to obtain slices from them with a reduced size. Image preprocessing has been applied to the CD2014 dataset using OpenCV [23]. This library allows us to resize the input images to dimensions multiple of our desired input size: 64×64. As a result, we generate blocks of 64×64 pixels from the three images: Reference, target and ground truth. This is achieved using the sliding window algorithm explained in Section 3.2. The resultant patches obtained from the previous process are automatically labelled using the original image name, category and region position to conform a unique identifier. With this implementation, the image order is preserved and training can be performed without any trouble.

### 3.5. Training

Our system has been implemented in Keras [28] using Tensorflow as backend [29]. As mentioned in previous sections, our model input is the result of the concatenation of the reference and the foreground image along the depth axis. The model output is conformed by a vector with a length of 4096 elements. To obtain this desired structure, we have to process our original ground truth images. These images are provided by the CD2014 Dataset as a grayscale image. Black pixels represent unmodified regions. White pixels depict altered zones and grey indicate the border between an altered and an unaltered zone. The image processing has been performed using the ImageDataGenerator class from Keras and Numpy [30]. ImageDataGenerator allows Keras to select batches of data for training the model, instead of loading the complete dataset on memory and select the batches from memory. As a result, we obtain an iterative component called generator. These generator structures are widely implemented in Keras, and training methods are no exception. With this methodology, extensive datasets are easy to handle. These generators can be easily customized. In our case, we have performed two significant customizations. First, we have concatenated both input images into a unique structure with size of 64×64×6 as our model input. Then, output images are transformed into vectors using Numpy API. Final dataset is detailed on Table 3. In summary, the training dataset is conformed by 130,476 patches, 43,492 for background, foreground and ground-truth respectively, with a proportion of 80% for training and 20% for evaluation purposes. 6750 additional patches from the IOM category have been included for metrics evaluation. Training loss is illustrated in Figure 5. The X-axis represents the steps (in thousands) of training. The Y-axis depict the loss value at a given step. The graph has been obtained using Tensorboard from TensorFlow. Tensorboard has been employed to analyse the training process. The model has been trained on an NVIDIA 1080TI GPU for 12 epochs because of the implementation of early stopping [31].

### 3.6. Post Processing

After the deep learning model is applied to both images, the resultant grayscale pictures are processed. As stated before, the system outputs a set of 64×64 grayscale images. The length of the mentioned set is proportional to the image dimensions. It is a result of the sliding window algorithm applied to the inputs, as detailed in Section 3.2. If overlapping has been selected in the sliding window algorithm, the post-processing method divides the obtained values at the composition process by the overlapping factor. With this approach, the overlapping effect allows us to have a trade-off between precision and computational cost, to obtain a more flexible implementation. Therefore, the image is obtained by inverting the sliding window algorithm. That is to say, the 64×64 patches are placed on the equivalent position of the input patch at the original image in a new blank image. After all the patches have been included in the new image, we obtain a grayscale image with the dimensions of the inputs of the system. This image will be introduced into the filtering component of this post-processing module. Figure 6 represents an example output previous to the filtering component.

Pixel’s intensity threshold is applied to filter possible noise effects such as blocking or insignificant changes detection in the image. Subsequently, a morphological dilation is performed on the resultant image. The objective of the dilation is to complete the possible gaps produced by noise in changed regions. Dilation has been selected as some information is typically removed by the filter to secure the complete elimination of disturbances. To compensate this, the dilation process expands the borders of the most relevant regions to complete the missed information after the previous threshold process. The dilation operator follows the formula:(8)$$\left(f⊕b\right)\left(x\right)=\underset{y\epsilon E}{\sup}\left[f\left(y\right)+b\left(x-y\right)\right]$$
denoting by *f*(*x*) an image, and the structuring function by *b*(*x*). *E* is the Euclidean space into the set of real numbers.

The effect of this filtering on the images is depicted in Figure 7.

## 4. Experimental Results

Model results are described in Section 4.1 along with the metrics used to analyse the model’s performance. In Section 4.2, we have compared our solution with several state-of-the-art implementations for background subtraction and change detection, trained using the CD2014 dataset.

### 4.1. Evaluation Metrics

To perform a comparison between our system and several state-of-the-art methods, we have selected multiple metrics to characterize the performance of the models. All of them are based on four elemental concepts. We define true positives (TP) as the correctly classified changed pixel. True negatives (TN) represent the unmodified pixels which have been correctly predicted. “Let false positives” (FP) defines the incorrect classified changed pixels. Lastly, false negatives (FN) represent the incorrectly labelled background pixels. We have selected the metrics from [1], as all the methods based on this dataset compute them. These metrics, defined by the previous concepts, are:Recall (Re): $\frac{TP}{TP+TN}$Specificity (Sp): $\frac{TP}{TP+TN}$False Positive Rate (FPR): $\frac{FP}{FP+TN}$False Negative Rate (FNR): $\frac{FN}{TP+FN}$Percentage of Wrong Classifications (PWC):$100*\frac{FP+FN}{TP+TN+FN+FP}$Precision (Pr): $\frac{TP}{TP+FP}$F-measure (FM): $2*\frac{Re*Pr}{Re+Pr}$

In Figure 8 we have represented examples of the model’s performance in different image categories. Similarly, Figure 9 depicts examples of our model’s performance in UAV imagery. Ultimately, Table 4 shows the results of the stated metrics represented for each scenario of our dataset. It should be noted that recall and precision measures are included in F-measure. From the metrics obtained, we can observe that the system has obtained excellent scores in complex scenarios such as snow blizzard.

### 4.2. Comparison with Other Change Detection Systems

As mentioned in the introductory part of this section, we have selected various implementations to compare our system. We have based our election in the use of the CD2014 dataset to have the most objective results as possible. Moreover, we have elected algorithms founded on various technologies such as traditional image processing techniques (SuBSENSE [8]), other traditional mathematical implementations (GMM [6]) and convolutional neural networks (ConvNet-IUTIS, ConvNet-GT [4]). Our purpose with this is to remark the performance of CNNs over traditional methods. As we compare a CNN based implementation, we can obtain valuable conclusions for our system performance. The mentioned traditional solutions have been selected because of their impact in background subtraction, as they are frequently compared in most papers dealing with this subject. In Table 5, the results of this comparison are represented, using the value of the F-measure as a reference for their performance. Only scenarios that have been tested with the model are compared. From the previous table, we can observe how the performance of our model denotes higher precision than other state-of-the-art implementations for static cameras. Therefore, our change detection component offers an accurate solution for the purpose. The combination of this development with the image alignment methods provides a state-of-the-art solution for remaking modified regions in images acquired by a moving camera.

## 5. Discussion

The primary purpose of this paper is to describe the development of a new system for change detection with UAV images using a combination of image alignment and CNNs. The most significant difference between our proposed solution and other state-of-the-art methods is the inclusion of the mentioned image alignment system to adjust images from moving cameras. Figure 10 depicts the improvement of the precision of our system caused by the use of the image alignment component. An additional difference in our system is represented by the improvement of precision it provides on the studied scenarios, as reflected in Table 5. Based on these reasons, we can affirm that our CNN architecture and training have been implemented effectively.

As mentioned before, the system employs a reference sequence to detect the changes. This implies restraints to the method as the reference must denote the ideal status of the recorded scenario. That is to say, all the elements included in this sequence are considered as background. Previous to this approach, a reconstruction method similar to [2] was implemented. Because of the unsatisfactory precision obtained and the computational cost involved, the approach was discarded.

An additional point to discuss would be the multiple variations that could appear between two UAV flights. The effect of flying at different heights is one of these possible variations that could occur on a real scenario. In this case, our system compensates this using the image alignment component to select the most relevant elements which are included on both reference and foreground images. At that point, it automatically selects the central region of both images to be analysed.

As can be observed in Table 5, implementations from [4] have an unknown F-measure score at the “Intermittent Object Motion” category. These methods do not consider this particular type of scenario. However, the results they obtain on the other two categories position them as a relevant state-of-the-art implementation to compare our system.

## 6. Conclusions

In this paper we have presented a change detection system for static and moving cameras using image alignment based on ORB algorithm and convolutional neural networks. Because of the use of UAV imagery acquired by moving cameras, the problem of dynamic backgrounds has been addressed. As we have detailed along the paper, our mayor improvement from other state-of-the-art implementations consists on the use of an image alignment process. The objective of this element is to compensate the possible variations during UAVs flights described in previous sections. In addition to that, the inclusion of the sliding window algorithm reduces the computational cost of the CNN model by reducing the dimensions of the input images. Moreover, this method adds versatility to the system. The sliding window algorithm can be adjusted to provide overlapping sections to improve accuracy with an increment in computational cost. As far as we know, a moving camera scenario has not been taken into account in any of the state-of-the-art methods for change detection compared along the paper. Only dynamic backgrounds on static cameras have been studied on mentioned implementations. Our system is capable of adapting to these conditions using image alignment techniques and the idea of a reference video or image. Datasets with dynamic backgrounds have been selected to train the network to achieve meaningful outcomes for real-world applications. Results from experiments indicate a precise detection in scenarios with adverse weather such as a snowfall or a blizzard. The comparison with other state-of-the-art methods reflects that our system is the most accurate on the studied scenarios.

### Future Work

The precision of the reference’s acquisition is crucial for the system’s performance. As a solution to this, we are working on GPS data processing for improving the alignment system as in [22]. In addition to that, we consider the option to include deep learning in the alignment process as another of our futures lines of work. Our objective with that is to compare the performance of deep learning against our current image alignment system based on ORB.

## Figures and Tables

**Figure 1 sensors-19-04484-f001:**
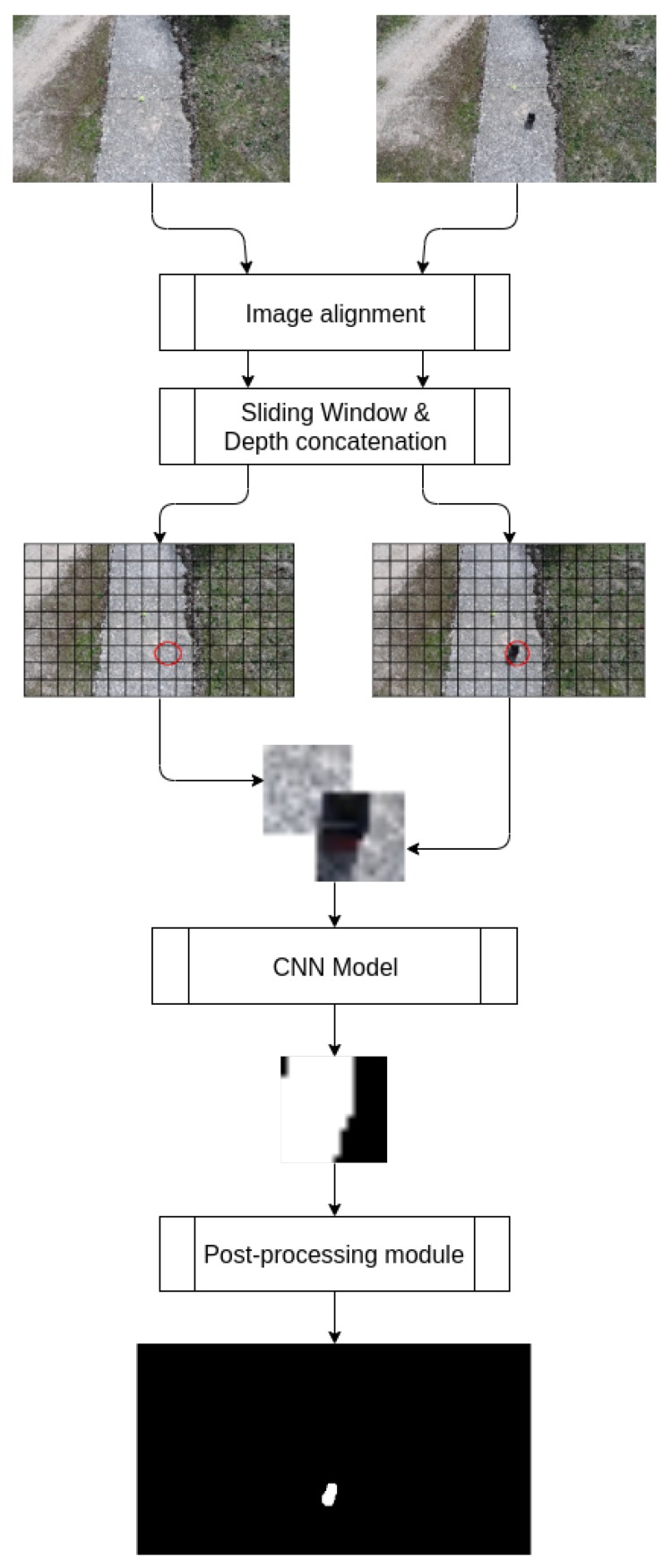
Block diagram of the system. Both reference and foreground images are introduced into our image alignment system. After that, sliding window algorithm is applied. The two resultant patches are concatenated along the depth axis. Ultimately, our CNN model predicts a grayscale image, which is post processed to obtain the final binary patch depicted.

**Figure 2 sensors-19-04484-f002:**
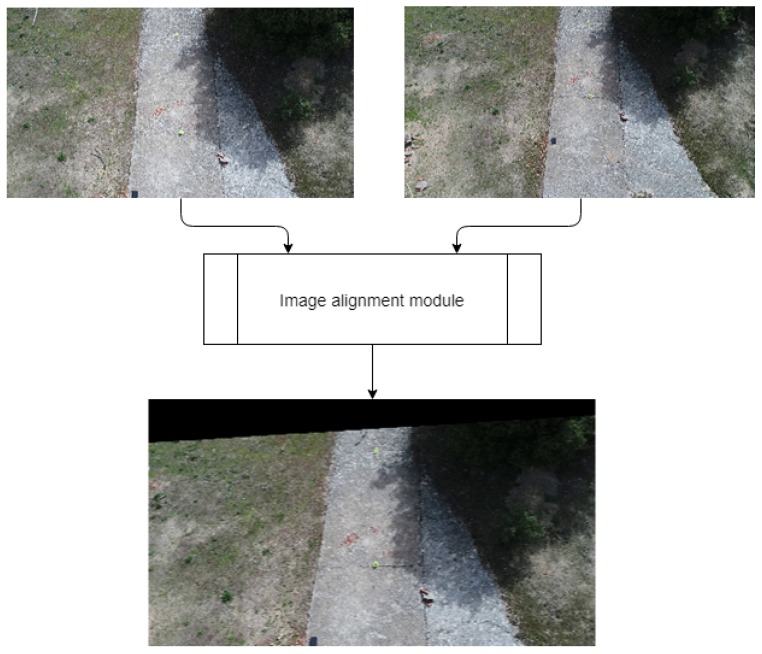
Example of our image alignment module’s output with a foreground image with significant difference from the reference.

**Figure 3 sensors-19-04484-f003:**
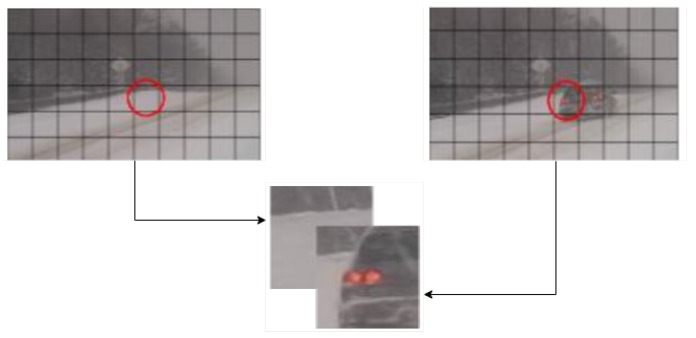
Representation of the sliding window and depth concatenation process on a reference image and a foreground image.

**Figure 4 sensors-19-04484-f004:**
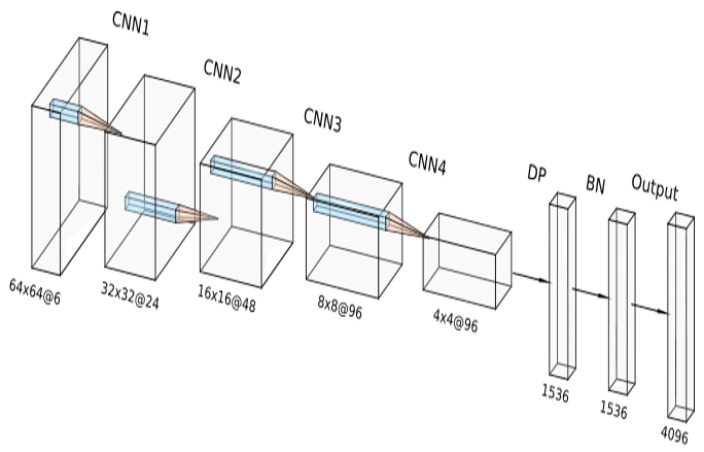
Diagram of the deep neural network architecture used in our system, containing layer types and dimensions.

**Figure 5 sensors-19-04484-f005:**
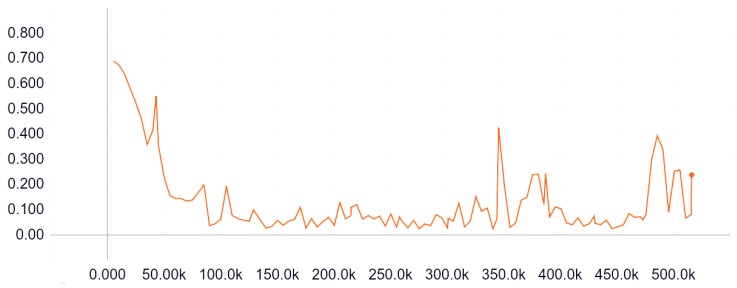
Training loss curve represented loss value per thousand steps.

**Figure 6 sensors-19-04484-f006:**
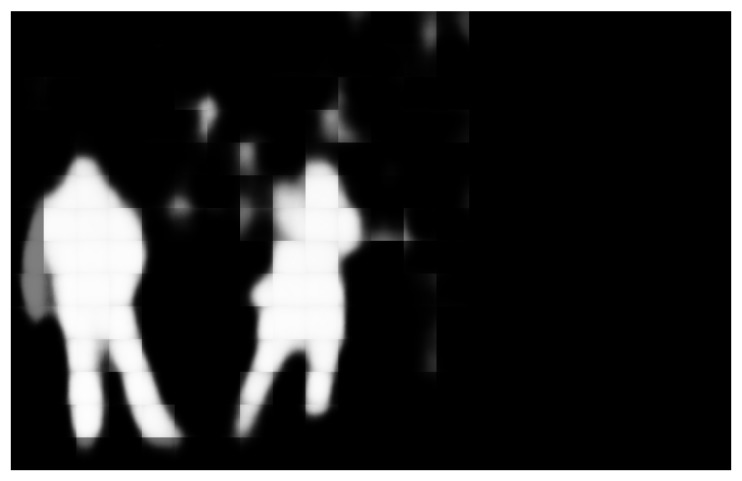
Image resultant of the combination of 64×64 grayscale patches obtained by CNN predictions.

**Figure 7 sensors-19-04484-f007:**
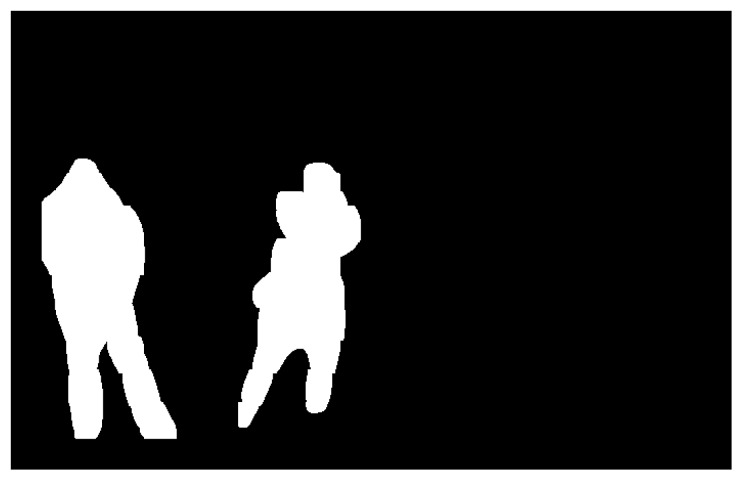
Final output of our system after the filtering part of our post-processing module is applied to Figure 6.

**Figure 8 sensors-19-04484-f008:**
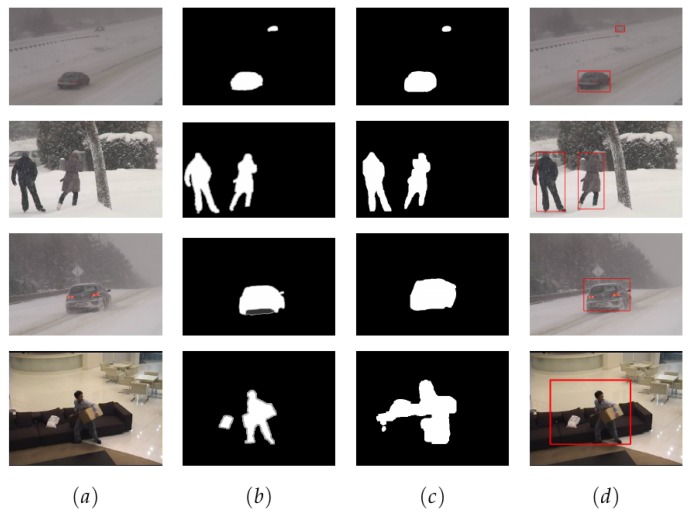
Results of our system applied to CD2014 images. Column (**a**) represents the image where change detection is applied, column (**b**) represents the ground truth images given by the CD2014 dataset, (**c**) the prediction images from our model and (**d**) the original images with the bounding boxes obtained by the prediction process.

**Figure 9 sensors-19-04484-f009:**
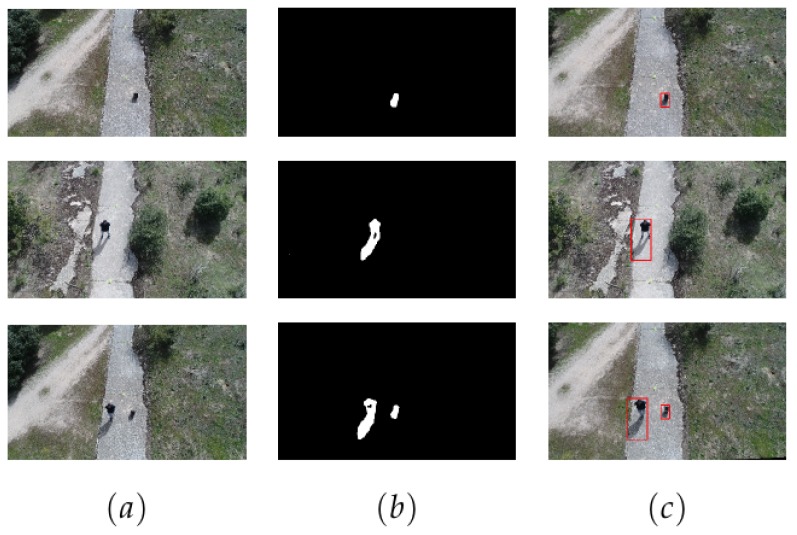
Results of our system applied to images acquired from an unmanned aerial vehicle (UAV), not included in the dataset. In this case, column (**a**) represents the foreground image, column (**b**) the binary image predicted and column (**c**) the original image bounding boxes as in previous table.

**Figure 10 sensors-19-04484-f010:**
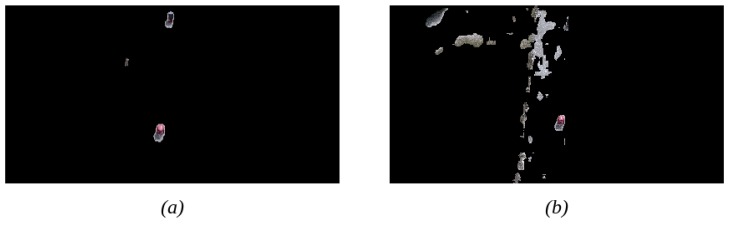
Effect of the image alignment component on the system’s output. (**a**) The resultant image with the image alignment applied, while (**b**) represent the results obtained without the use of the component. As can be observed, the aligned image detects changed elements (in this case, the two red boxes) more precisely.

**Table 1 sensors-19-04484-t001:** Layer description of the convolutional neural networks (CNNs) which conform the first part our model.

Name	Input Size	Kernel	Stride	Output Size
conv2d1	64 × 64 × 6	3 × 3	1	64 × 64 × 24
pool1	64 × 64 × 24	-	2	32 × 32 × 24
conv2d2	32 × 32 × 24	3 × 3	1	32 × 32 × 48
pool2	32 × 32 × 48	-	2	16 × 16 × 48
conv2d3	16 × 16 × 48	3 × 3	1	16 × 16 × 96
pool3	16 × 16 × 96	-	2	8 × 8 × 96
conv2d4	8 × 8 × 96	3 × 3	1	8 × 8 × 96
pool4	8 × 8 × 96	-	2	4 × 4 × 96

**Table 2 sensors-19-04484-t002:** Layer description of the fully connected layers which define the final part of our model.

Name	Input Size	Function	Output Size
flatten	4 × 4 × 96	-	1536
bn	1536	Batch Normalization	1536
dp	1536	Dropout	1536
final	1536	Sigmoid	4096

**Table 3 sensors-19-04484-t003:** Description of the complete dataset used to train our model. Columns represent the category of the images, the number of complete images used, the size of each image and the resultant number of 64×64 patches used to train the algorithm.

Category	Images	Size	Total Patches
Blizzard (BW)	195	1216 × 704	40,755
Skating (BW)	177	1216 × 704	36,993
Snowfall (BW)	192	1216 × 704	40,128
Canoe (DBG)	210	640 × 384	12,600
Sofa (IOM) ^1^	150	320 × 192	6750

^1^ Only used for metrics evaluation.

**Table 4 sensors-19-04484-t004:** Metrics scores of our proposed solution on each CD2014 selected category.

Category	Sp	FPR	FNR	PWC	FM
Bad Weather (BW)	0.999914	0.000086	0.022035	0.017216	0.977963
Dynamic Background (DBG)	0.999810	0.000190	0.048662	0.038013	0.951339
Intermittent Object Motion (IOM)	0.999790	0.000210	0.005232	0.004808	0.994768

**Table 5 sensors-19-04484-t005:** F-measure scores of our implementation and the other seven state-of-the-art implementations on the CD2014 dataset from categories “Bad Weather”, “Dynamic Background” and “Intermittent Object Motion”.

Implementation	FM_overall_	FM_BW_	FM_DBG_	FM_IOM_
Proposed	**0.9747**	**0.9779**	**0.9513**	**0.9947**
ConvNet-GT [4]	0.9054	0.9264	0.8845	Unknown
ConvNet-IUTIS [4]	0.8386	0.8849	0.7923	Unknown
CNN [3]	0.7718	0.8301	0.8761	0.6098
GMM [6]	0.6306	0.7380	0.6330	0.5207
SuBSENSE [8]	0,7788	0.8619	0.8177	0.6569
PBAS [32]	0.6749	0.7673	0.6829	0.5745
PAWCS [5]	0.8285	0.8152	0.8938	0.7764

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
