# Peer review of "Automatic Change Detection System over Unmanned Aerial Vehicle Video Sequences Based on Convolutional Neural Networks"

_sensors, 2019, doi:10.3390/s19204484_

Round 1

Reviewer 1 Report

General comments:

* vARGOS doesn't sound like an abbreviation. Wonder how the authors came up with the term.
* The article lacks details on methods on which the work is based on.
* Missing references to similar works, which can be found just by searching in Google scholar.
* It would be interesting to see the results on how good the image alignment/registration works.
* Could improve the structure, flow, and presentation.
* Many Figures can have descriptive captions
* Missing discussion
* Need some language and grammar corrections.

Section-wise comments:
Abstract:
* Could clarify what does it mean by 'variations on the UAV flight', changes in the environment? What kinds of different environments?
* Not a good idea to cite in the abstract.

1.
* The authors have used the terms UAV and drones without making it clear whether they are using them interchangeably. Or, otherwise could differentiate between them.
* I would expect some references to back up statements like 'we have observed an increasing tendency to use CNNs on image processing systems for change detection. Furthermore, most of the studied implementations do not consider a moving camera'.
* CNN is used in different applications and for different purposes. It is not clear for what purpose from the statement 'we have created our own complete approach based on image alignment and CNNs'. Suggest using other more appropriate terms than 'create'
* The whole introduction section seems to me a bit short and is a single paragraph. This could be divided into several paragraphs. And, maybe better to describe in a bit more details about the problem, current status, and issues towards addressing the problem, etc.

2.

* It is worth elaborating more on the methods referred in 2.1, how these methods detect change.
* I do not think there is a need for a long explanation to CNN (2.2). Rather, it would be good to describe more on how it is used in the context of the problem at hand (2.3).
* Would be good to talk about what kind of automation/innovation and which systems as referred to [12] in section 2.4.
* Any specific reason for using ORB for image alignment/registration. It is well known that these approaches are faster but not so robust compared to SIFT and SURF based methods.

3.
* I'd expect a description of the methods used, at least in brief, not just refer to other papers. For example, descriptor matcher referred to [20].
* Missing details on how sliding window and depth concatenation works in Figure 1. Which CNN model is it?
* What does it mean by 'ReLU activation applies 5 to its inputs' in 3.2?
* Could elaborate how the authors came up with the DCNN architecture.
* Figure 2. can possibly be removed, as it is well known to the community.
* The image resolution 64x64 used seems to be quite low.
* How many IOM images? Missing in Table 3?
* Not sure how the image numbers 130476, 43492 came and what are total patches in Table 4.
* Wonder if you considered/tried with black and white output image instead of grayscale and post-processing the grayscale image to black and white image.

4.
* Instead of Table 5, it can be a figure. And, could describe it with bit detailed caption. Same for Table 6.
* Could briefly introduce different methods used (in related works) in the comparison.
* Wonder why two results are 'unknown' for FM_IOM in Table 7.

5.
* Since comparative results have been given with DBG images from other state of the art methods as well, I had difficulty understanding the claim that DBG issue has not been taken into account by other methods. And, apart from training the model with such ground truths, I probably missed the point on how the issue has been addressed specifically in the proposed method.

Reviewer 2 Report

The paper was on an interesting topic, but there were several flaws. First the writing (grammar, English style) needs improvement. Second, the structure of the paper mixes methods, results and discussion. This may be ok if it is done well, but this brings up the Third point, that there is not proper analysis of the results, no discussion other than to say that the authors' results are best. Deeper description comparing the results from the authors' system with those from other systems is appropriate. How do they differ? Are these critical differences? 

I would also like to know (or see emphasized more clearly) what the differences are between the authors's system and the code already written by others. It is mentioned at some points what the advantages are of the vARGOS system, but this should be lifted as it's rather lost in the text now. 

A figure showing a schematic of the system (input, processes, output) would also be helpful. 

Specific comments for when you will improve the paper:

Major English corrections, such as changing expressions and grammar, are needed,.

Abstract should include quantification of results.

Ln 58 – Change detection is much more than this. “Traditional change detection” from a remote sensing perspective, is what you are referring to and there is much more and fundamental literature you should refer to if you want to give a background on Change Detection in remote sensing. Narrow your approach to the type of Change Detection you may be referring to (eg in surveillance, or from rapidly changing images). References 5 and 6 are very specific change detection approaches – so please make it clear that your paper is very specific rather than give the feeling that a broad review is being done.

Ln69 – Give an actual short description of CNNs rather than start with referring the reader to another paper.

Ln 98 – VGG?

Section 2.4 – “…is assumed to vary from two drone flights…” Vary how?  Be specific

Ln 120 – Define vARGOS before using acronym. This is the first mention.

Ln 122 – So a reference background image is needed. Please mention (in the appropriate section which may be in the Discussion) what happens if something in the reference image changes, and how what is “permanent” vs “a change” in the initial reference image is determined.  Please discuss effect of flying at different heights (and therefore comparing images of different scales) – in other words, is it a requirement of your system that the scale of images (reference and change) are exactly the same?

Section 3.5 was particularly confusing, but here the schematic could help.

Table text is needed. Description of Figure and Tables need improvement so they stand on their own. Tables 5 and 6 are really Figures. 

The Conclusion takes up several points never mentioned in the paper itself. This shouldn't be the case. The very last sentence I cannot interpret.

Round 2

Reviewer 2 Report

Thanks for the changes.

Starting out the Introduction with "These methods..." doesn't work. Be specific about your paper. 

You still need corrections to the English of the paper by a native speaker. There are missing words, awkward phrasing, text that doesn't make sense, etc. It detracts from the content of the paper, and to a native English speaker, is almost not readable. What is the standard of the journal? 

What is the ? on line 132?

Take away from Fig 4 and Table 1 and 2 text the word "Detailed"

Fig 1 - Can you improve the bottom image and indicate it as your final output.
